# Indiscriminate Poisoning Attacks on Unsupervised Contrastive Learning

**Hao He**[*]**, Kaiwen Zha**[*]**, Dina Katabi**
Computer Science and Artificial Intelligence Lab
Massachusetts Institute of Technology
{haohe,kzha,dk}@mit.edu

## Abstract

Indiscriminate data poisoning attacks are quite effective against supervised learning. However, not much is known about their impact on unsupervised contrastive learning (CL). This paper is the first to consider indiscriminate poisoning attacks of contrastive learning. We propose *Contrastive Poisoning (CP)*, the first effective such attack on CL. We empirically show that Contrastive Poisoning, not only drastically reduces the performance of CL algorithms, but also attacks supervised learning models, making it the most generalizable indiscriminate poisoning attack. We also show that CL algorithms with a momentum encoder are more robust to indiscriminate poisoning, and propose a new countermeasure based on matrix completion. Code is available at: `https://github.com/kaiwenzha/contrastive-poisoning`.

## 1 Introduction

Indiscriminate poisoning attacks are a particular type of data poisoning in which the attacker adds to the training data or labels perturbations that do not target a particular class, but lead to arbitrarily bad accuracy on unseen test data. They are also known as availability attacks (Biggio & Roli, 2018) since they render machine learning models potentially useless, or delusive attacks (Tao et al., 2021) since the added perturbations are visually imperceptible.

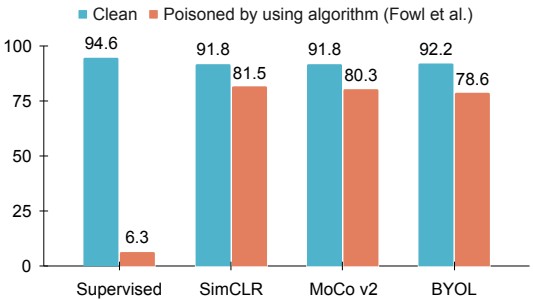

Figure 1: Accuracy of the victim model when facing the current SOTA in indiscriminate data poisoning attacks (Fowl et al., 2021a). It shows that past indiscriminate poisoning while highly effective on victim models that use supervised learning is mostly ineffective when the victim uses unsupervised contrastive learning (SimCLR, MoCo, BYOL). Experiment is on CIFAR-10.

Research on indiscriminate poisoning attacks has attracted much attention in recent years due to concerns about unauthorized or even illegal exploitation of online personal data (Prabhu & Birhane, 2021; Carlini et al., 2021). One example is reported by Hill & Krolik where a commercial company collected billions of face images to build their face recognition model without acquiring any consent. Indiscriminate poisoning attacks could protect from such unauthorized use of data because the added poison renders models trained on the data ineffective (Fowl et al., 2021b).

All prior works on indiscriminate poisoning of deep learning are in the context of *supervised learning (SL)*, and use a cross-entropy loss. However, advances in modern machine learning have shown that unsupervised *contrastive learning* (CL) can achieve the same accuracy or even exceed the performance of supervised learning on core machine learning tasks (Azizi et al., 2021; Radford et al., 2021; Chen et al., 2020b; 2021; Tian et al., 2021; Jaiswal et al., 2021). Hence, an individual or a company that wants to use a dataset in an unauthorized manner need not use SL. Such a malicious company can use CL to learn a highly powerful representation using unauthorized data access.

---

[*]Equal contribution, determined via a random coin flip

This paper studies indiscriminate data poisoning attacks on CL. We observe that past works on indiscriminate poisoning attacks on supervised learning do not work in the face of contrastive learning. In Figure 1, we show empirically that three popular contrastive learning algorithms, SimCLR (Chen et al., 2020a), MoCo (Chen et al., 2020c), BYOL (Grill et al., 2020) are still able to learn highly discriminative features from a dataset poisoned using a state-of-the-art indiscriminate SL poisoning (Fowl et al., 2021a). In contrast, the attack renders supervised learning completely ineffective. This is because indiscriminate poisoning of SL generates poisoning perturbations that are clustered according to the class labels (Yu et al., 2022); but such a design is unlikely to be effective against unsupervised CL because its representation learning does not involve any class labels.

We propose *Contrastive Poisoning (CP)*, the first indiscriminate data poisoning attack that is effective against CL. The design of CP involves three components: 1) a poison generation process that attacks the contrastive learning objective (e.g., the InfoNCE loss); 2) a differentiation procedure that attacks data augmentation in CL; and 3) a dual-branch gradient propagation scheme that attacks CL algorithms with a momentum encoder. We empirically evaluate CP on multiple datasets commonly used in prior work on indiscriminate poisoning attacks (CIFAR-10/-100, STL-10, and ImageNet-100). Our results reveal important new findings:

- Contrastive learning is more robust to indiscriminate data poisoning than supervised learning. All prior indiscriminate poisoning attacks on SL can be evaded by using CL to learn a representation, then use the labels to learn a linear classifier.

- Contrastive Poisoning (CP) is a highly effective attack. Interestingly, the same contrastive poison can attack various CL algorithms (SimCLR, MoCo, BYOL), as well as supervised learning models.

- While CP works on all CL algorithms, models that include a momentum encoder (i.e., MoCo and BYOL) are relatively more robust than those that do not (i.e., SimCLR). Furthermore, it is essential to generate the poison using a momentum encoder; poisons generated without a momentum encoder, i.e., using SimCLR, do not generalize well to other CL algorithms.

- The best defenses against poisoning SL (i.e., adversarial training (Tao et al., 2021)) cannot defend against contrastive poisoning. A new data augmentation based on matrix completion works better.

## 2 RELATED WORK

**Indiscriminate poisoning attacks.** Indiscriminate poisoning attacks have been well studied in the context of classical machine learning models, like linear regression and support vector machine (Barreno et al., 2006; Biggio et al., 2012). Indiscriminate poisoning attacks on deep neural networks have recently become a trendy topic due to the need for protecting data from unauthorized use (Muñoz-González et al., 2017; Feng et al., 2019; Shen et al., 2019; Shan et al., 2020; Cherepanova et al., 2021; Yuan & Wu, 2021; Huang et al., 2021; Fowl et al., 2021a;b).

All prior work on indiscriminate data poisoning of deep learning targets supervised learning and uses a cross-entropy loss. The closest past work to ours is in the area of targeted poisoning and backdoor attacks on contrastive learning (Carlini & Terzis, 2022; Jia et al., 2022; Liu et al., 2022). Targeted poisoning attacks perturb the training data to make the poisoned model misclassify a specific data sample (as opposed to all unseen data). Backdoor poisoning attacks, on the other hand, implant a backdoor into the poisoned model to manipulate its behavior only on inputs that include the backdoor trigger (as opposed to any clean input). Carlini & Terzis investigates targeted attacks and backdoor attacks on a specific multi-modality contrastive learning framework called CLIP (Radford et al., 2021). Saha et al. mounts a backdoor attack on contrastive learning by adding triggers to all images from one class in the training set. Truong et al. uses the contrastive loss as a regularization to make neural networks more resilient to backdoor attacks. Our work differs from all of the above attacks and is the first to focus on indiscriminate poisoning of contrastive learning.

**Indiscriminate poisoning defenses.** Past studies (Tao et al., 2021; Huang et al., 2021; Fowl et al., 2021a; Geiping et al., 2021) have shown that adversarial training (Madry et al., 2018) is the most effective way to counter indiscriminate poisoning attacks. They also considered other defense mechanisms such as protecting the learning process by using differentially-private optimizers like DP-SGD (Hong et al., 2020), and data augmentation techniques (Borgnia et al., 2021; Fowl et al., 2021a) such as additive noise and Gaussian smoothing, Cutout (DeVries & Taylor, 2017), Mixup (Zhang et al., 2018), and CutMix (Yun et al., 2019). This past work is in the context of SL. No past work has investigated defenses against indiscriminate poisoning attacks on CL.

## 3  THREAT MODEL

**Attacker objective.** We consider the standard setting, where contrastive learning is used to learn a feature extractor in a self-supervised manner without labels (Chen et al., 2020a). The feature extractor is then fixed, and used to train a predictive head on some downstream task of interest.

The ultimate goal of our attack is to poison the training set to cause the contrastive model learned by the victim to become a poor feature extractor. The performance of the feature extractor is evaluated on a downstream task with a task-specific predictor. We focus on the setting where the predictor is linear. This evaluation approach is known as *linear probes* (Alain & Bengio, 2017).

**Attacker capability.** In the context of indiscriminate poisoning attacks, the attacker is usually assumed to have the ability to access the training data of the victim and poison this dataset to degrade the performance of the victim's learned model. The attacker however cannot interfere with the victim's architecture choice, model initialization, and training routines.

The victim may use one of the common CL algorithms (e.g., SimCLR, MoCo, BYOL). In our study, we consider both cases of the victim's algorithm being known or unknown to the attacker. As in past work on indiscriminate poisoning (Yu et al., 2022), the attacker is allowed to modify a large portion of the clean training samples. By default, we poison $100\%$ of the data in our experiments. However, the attacker is constrained to only perturb the data samples without touching the labels, and the perturbation should be imperceptible. We follow the convention of prior works which allows the attacker to perturb the data in an $\ell_\infty$ ball with $\epsilon = 8/255$ radius.

**Notations.** We use $\mathcal{D}$ for the dataset, $\mathcal{X}$ and $\mathcal{Y}$ for the data and label space. We consider classification tasks with $C$ classes, i.e., $\mathcal{Y} = [C]$; $h : \mathcal{X} \to \Delta_\mathcal{Y}$ denotes a classifier, which is a composition of a feature extractor $f : \mathcal{X} \to \mathbb{R}^d$ and linear predictor $g : \mathbb{R}^d \to \Delta_\mathcal{Y}$, where $\Delta_\mathcal{Y}$ is the probability simplex; $l$ denotes a loss function; $\mathcal{L}(h; \mathcal{D})$ is the averaged loss of a model $h$ on the dataset $\mathcal{D}$. We use $\mathcal{L}_{\text{CE}}(h; \mathcal{D})$ to refer to the cross-entropy loss used to learn a classifier $h$. We use $\mathcal{L}_{\text{CL}}(f; \mathcal{D})$ to refer to the contrastive loss (e.g. InfoNCE) used to learn a feature extractor $f$.

We formalize indiscriminate poisoning attacks on unsupervised contrastive learning as follows. First, the attacker has a clean version of the training data $\mathcal{D}_c$ and generates a poisoned version $\mathcal{D}_p$. The victim applies a certain contrastive learning algorithm to $\mathcal{D}_p$, and obtains a poisoned feature extractor $f_p = \arg\min_f \mathcal{L}_{\text{CL}}(f; \mathcal{D}_p)$. To evaluate its goodness, we employ a new labeled downstream dataset $\mathcal{D}_e$, and train a linear predictor $g_p = \arg\min_g \mathcal{L}_{\text{CE}}(g \circ f_p; \mathcal{D}_e)$. The accuracy of the resulting classifier $h_p = g_p \circ f_p$ on the downstream dataset $\mathcal{D}_e$ is used to assess the effectiveness of the attack.

## 4  CONTRASTIVE POISONING

We propose a new indiscriminate data poisoning attack, *Contrastive Poisoning (CP)*, that disables CL algorithms from learning informative representations from training data. The design of CP has three components that together deliver a highly potent attack, which we describe below.

### 4.1  CONTRASTIVE POISON GENERATION

CL algorithms (SimCLR, MoCo, BYOL) contain a feature encoder and a predictor. The idea underlying our poison generation is to learn data perturbations that, when added to a clean dataset, fool the feature extractor of the victim and make it optimize the contrastive learning objective without really learning semantic information. To achieve this goal, the attacker first chooses a target CL algorithm (like SimCLR or MoCo) and co-optimizes both the feature extractor $f_\theta$ (with parameter $\theta$) and the perturbations $\delta(x)$. Specifically, we solve the following optimization:

$$\min_{\theta, \delta : \|\delta(x)\|_\infty \leq \epsilon} \mathbb{E}_{\{x_i\}_{i=1}^B \sim \mathcal{D}_c} \mathcal{L}_{\text{CL}}(f_\theta; \{x_i + \delta(x_i)\}_{i=1}^B). \tag{1}$$

where $\mathcal{L}_{\text{CL}}(f_\theta; \{x_i + \delta(x_i)\}_{i=1}^B)$ is the CL loss (e.g., InfoNCE) computed on a batch of samples. Our algorithm solves the above optimization by alternating between updating the feature extractor for $T_\theta$ steps and updating the poisoning noise for $T_\delta$ steps. The feature extractor is optimized via (stochastic) gradient descent while the poisoning perturbation is optimized via projected gradient descent (Madry et al., 2018) to fulfill the requirement of a bounded $\ell_\infty$ norm.

---

**Algorithm 1** Contrastive Poisoning

---

1: **Input:** clean dataset $\mathcal{D}_c$; learning rate $\eta_\theta, \eta_\delta$; number of total rounds $T$; number of updates in each round $T_\theta, T_\delta$ for the feature extractor, poisoning perturbations; number of PGD steps $T_p$.
2: **for** outer iteration= $1,\ldots,$T **do**
3:     **for** inner iteration= $1, 2, \ldots, T_\theta$ **do**
4:         Sample a batch of data $\{x_i\}_{i=1}^B \sim \mathcal{D}_c$
5:         $\theta \leftarrow \theta - \eta_\theta \nabla_\theta \mathcal{L}_{\text{CL}}(f_\theta; \{x_i + \delta_i\}_{i=1}^B)$
6:     **end for**
7:     **for** inner iteration= $1, 2, \ldots, T_\delta$ **do**
8:         Sample a batch of data $\{(x_i, y_i)\}_{i=1}^B \sim \mathcal{D}_c$
9:         **for** iteration= $1, 2, \ldots, T_p$ **do**
10:             Compute gradient $g_i \leftarrow \nabla_{\delta_i} \mathcal{L}_{\text{CL}}(f_\theta; \{x_i + \delta_i\}_{i=1}^B)$
11:             $\delta(x_i) \leftarrow \Pi_\epsilon \left(\delta(x_i) - \eta_\delta \cdot \text{sign}(g_i)\right)$         ▷ **for sample-wise poisoning**
12:             $\delta(y) \leftarrow \Pi_\epsilon \left(\delta(x_i) - \eta_\delta \cdot \text{sign}(\sum_{i:y_i=y} g_i)\right), \forall y$    ▷ **for class-wise poisoning**
13:         **end for**
14:     **end for**
15: **end for**
16: **Output:** poisoned dataset $\mathcal{D}_p = \{x + \delta(x) : x \in \mathcal{D}_c\}$.

---

**Sample-wise and class-wise perturbations.** We consider two variants of contrastive poisoning: (1) *sample-wise* poisoning, which adds specific noise to each image, and (2) *class-wise* poisoning, which adds the same poison to all samples in the same class. Intuitively sample-wise poisoning provides more capacity to attack, but has the downside of increased learning overhead and potentially being too optimized to a specific CL algorithm. In Section 5.3, we empirically study the efficacy and properties of the two types of poisoning attacks. Algorithm 1 illustrates the poison generation procedure for both attack types.

## 4.2 DUAL BRANCH POISON PROPAGATION

Optimizing the noise to poison supervised learning is relatively straightforward; we can get the gradient through the cross-entropy loss function, e.g., $\nabla_x l_{\text{CE}}(h(x), y)$. Contrastive learning algorithms are more complicated since the computation of loss requires contrasting multiple data samples, and potentially the use of momentum encoders. Figure 2 illustrates the three contrastive learning frameworks, SimCLR, MoCo, and BYOL. As we can see, MoCo and BYOL have a momentum encoder which is updated via an exponential moving average of the normal encoder. In standard contrastive training, the gradient does not flow back through the momentum encoder, as illustrated by the black arrows in Figure 2. However, we note that the gradient from the momentum encoder is indispensable for learning the poisoning noise. We propose to learn the noise via gradients from both branches of the encoder and momentum encoder (blue arrows in Figure 2). We call it a *dual branch scheme*. To compare, we call learning noise via the standard gradient flow, a *single branch scheme*. Later in Section 5.5b, we empirically show that the proposed dual branch scheme learns much stronger poisons for both MoCo and BYOL.

## 4.3 ATTACKING CL DATA AUGMENTATIONS

Supervised learning uses weak data augmentations or no augmentations. Hence some prior work on poisoning SL has ignored data augmentations. Contrastive learning however uses strong augmentations, which are critical to the success of CL. If ignored, these augmentations can render the poison ineffective.

Hence, to optimize the poisoning noise to attack CL augmentations, we have to optimize the poisoning perturbations by back-propagating the gradients through the augmented views to the input samples. Fortunately, all data augmentations used in CL algorithms are differentiable. For example, brightness adjustment is one augmentation where the output $O$ is the input image $I$ multiplied by a factor $r_b$. It is differentiable as $\frac{\partial O_{c,h,w}}{\partial I_{c',h',w'}} = r_b \mathbf{1}_{[c=c',h=h',w=w']}$ where $h, w, h', w'$ are the spatial coordinates and $c, c'$ are the RGB channel indices. In Appendix A, we show the differentiability of all CL data

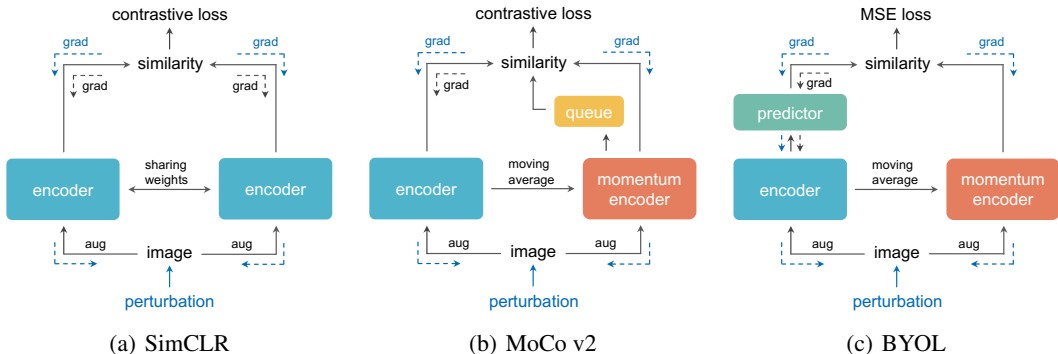

Figure 2: Contrastive learning frameworks and the gradient flow used to optimize the encoder (i.e., feature extractor) and the noise. The gradient flow for optimizing the encoder is shown in dashed black arrows and the flow for optimizing the noise in dashed blue arrows.

augmentations. Thus, to generate the contrastive poison we differentiate the data augmentation and back-propagate the gradients all the way to the input.

## 5 ATTACK EXPERIMENTS

We evaluate contrastive poisoning (CP) on multiple benchmark datasets: CIFAR-10/-100 (Krizhevsky et al., 2009), STL-10 (Coates et al., 2011), and ImageNet-100. ImageNet-100 is a randomly selected 100-class subset of the ImageNet ILSVRC-2012 dataset (Russakovsky et al., 2015), containing ∼131.7K images. As common in prior work (Huang et al., 2021; Fowl et al., 2021a), by default, we assume the victim uses ResNet-18, and the attacker generates the poisoning perturbations using ResNet-18. In Section 5.5c, we show ablation results that demonstrate that our data poisoning generalizes to other model architectures.

We consider three well-known contrastive learning frameworks: SimCLR (Chen et al., 2020a), MoCo(v2) (Chen et al., 2020c), and BYOL (Grill et al., 2020). For reference, in some experiments, we also evaluate two prior SOTA poisoning attacks against supervised learning: adversarial poisoning (AP) (Fowl et al., 2021a) and unlearnable example (UE) (Huang et al., 2021). In the rest of the paper, we use CP (S) and CP (C) as abbreviations for sample-wise and class-wise contrastive poisoning.

Please refer to Appendix B for details about the training and evaluation protocols, e.g., dataset split, hyper-parameters for CL frameworks and linear probing, as well as the hyper-parameters used to generate each attack.

### 5.1 CL ACCURACY IN THE FACE OF INDISCRIMINATE ATTACKS

We validate our attacks on CIFAR-10, CIFAR-100, and ImageNet-100. Here, we assume the attacker uses the same CL algorithm as the victim, and study attack transferability in Section 5.3. Table 1 reports the linear probing accuracy for different attack types against different victim CL algorithms. As a reference, in the first and second rows, we show the accuracy of training on clean data and clean data plus uniformly distributed random noise bounded by $[-8/255, 8/255]$, respectively.

The results in Table 1 show that both sample-wise and class-wise contrastive poisoning are highly effective in poisoning CL. They also show that CL algorithms that include a momentum encoder are less vulnerable than those that do not. For example, as illustrated by the bold numbers in Table 1, the strongest attack on CIFAR-10 causes SimCLR's accuracy to drop to 44.9%, whereas the strongest attacks on MoCo and BYOL cause the accuracy to drop to 55.1% and 56.9%, respectively. We believe the increased robustness of MoCo and BYOL is due to additional constraints on the poison, which not only has to make two views of an image similar, but also has to make them similar through two different branches (encoder and momentum encoder).

Table 1: Performance of indiscriminate poisoning attacks on different contrastive learning algorithms and datasets. Table reports percentage accuracy (%, ↓). For reference, we show the accuracy on clean data and clean data added random noise. The best attack performance is shown in **bold**.

| Attack Type | CIFAR-10 | | | CIFAR-100 | | | ImageNet-100 |
|---|---|---|---|---|---|---|---|
| | SimCLR | MoCo v2 | BYOL | SimCLR | MoCo v2 | BYOL | SimCLR |
| NONE | 91.8 | 91.8 | 92.2 | 63.6 | 65.2 | 65.3 | 69.3 |
| RANDOM NOISE | 90.4 | 90.1 | 90.7 | 58.5 | 59.8 | 61.0 | 67.5 |
| CONTRASTIVE POISONING (S) | **44.9** | **55.1** | 59.6 | **19.9** | **21.8** | 41.9 | **48.2** |
| CONTRASTIVE POISONING (C) | 68.0 | 61.9 | **56.9** | 34.7 | 41.9 | **39.2** | 55.6 |

Table 2: Impact of indiscriminate poisoning of one dataset on downstream tasks, where the linear predictor is learned using a different clean dataset. Table reports percentage accuracy (%, ↓) on features learned from poisoned CIFAR-10/ImageNet-100 with different downstream tasks. For reference, we also show the performance in the absence of attack.

| Attack Type | Poisoning on CIFAR-10 | | | Poisoning on ImageNet-100 | | |
|---|---|---|---|---|---|---|
| | CIFAR-10 | CIFAR-100 | STL-10 | ImageNet-100 | CIFAR-10 | STL-10 |
| NONE | 91.8 | 47.2 | 78.2 | 69.3 | 72.5 | 82.0 |
| CONTRASTIVE POISONING (S) | **44.9** | **16.7** | **43.1** | **48.2** | **59.9** | **67.8** |
| CONTRASTIVE POISONING (C) | 68.0 | 28.7 | 58.4 | 55.6 | 62.9 | 71.6 |

## 5.2 IMPACT ON DOWNSTREAM TASKS AND DATASETS

CL is widely-used for representation learning, where the goal is to learn a good feature extractor using unsupervised data. One can then apply the feature extractor to multiple downstream tasks by training a task-specific linear predictor on a new labeled dataset suitable for the downstream task.

The results in Table 2 show that after being poisoned, the feature extractor's discriminative ability on other datasets gets suppressed. Here, the victim uses SimCLR. We test the representations learned on two poisoned datasets, CIFAR-10 and ImageNet-100. The linear predictor is learned and tested on multiple datasets including, CIFAR-10/-100, STL-10, and ImageNet-100. As shown in Table 2, the accuracy on all downstream datasets drops unanimously, though the downstream datasets are clean and different from the poisoned dataset.

## 5.3 TRANSFERABILITY ACROSS CL ALGORITHMS

In practice, the attacker may not know which CL algorithm the victim uses. Thus, ideally, the attacker wants the poisoned data to be equally harmful regardless of which exact CL algorithm the victim uses. Thus, in this section, we assess the transferability of a particular attack across three potential victim CL algorithms (SimCLR, MoCo, BYOL). We conduct the experiments on CIFAR-10, and report the accuracy in Table 3. Bold indicates the most effective attack, and

Table 3: Accuracy for different victim's CL algs.

| Attack Type + Attacker's Alg. | Victim's Algorithm | | |
|---|---|---|---|
| | SimCLR | MoCo | BYOL |
| ADVERSARIAL POISONING | 81.5 | 80.3 | 78.6 |
| UNLEARNABLE EXAMPLE | 91.3 | 90.9 | 91.6 |
| CONTRASTIVE POISONING (S) (SIMCLR) | **44.9** | 82.0 | 85.4 |
| CONTRASTIVE POISONING (S) (MOCO) | 54.9 | **55.1** | 71.1 |
| CONTRASTIVE POISONING (S) (BYOL) | 65.1 | 64.2 | 59.6 |
| CONTRASTIVE POISONING (C) (SIMCLR) | 68.0 | 68.4 | 67.2 |
| CONTRASTIVE POISONING (C) (MOCO) | 60.9 | 61.9 | 59.5 |
| CONTRASTIVE POISONING (C) (BYOL) | 60.7 | 61.8 | **56.9** |

blue shading indicates the most transferable. For reference, in the first two rows, we also include the results of directly applying prior SOTA poisoning attacks against supervised learning.

The results show that prior poisoning attacks designed for supervised learning are very weak in countering contrastive learning, regardless of the victim's CL algorithm. In contrast, contrastive poisoning is highly effective. However, while sample-wise CP tends to be the most damaging, it does not transfer as well as class-wise CP. In terms of transferability Table 3 shows an attack that uses BYOL and class-wise CP provides the best tradeoff between transferability and efficacy, and causes the accuracy on CIFAR-10 to drop to about 60%, regardless of whether the victim uses SimCLR, MoCo, or BYOL.

## 5.4 POISONING BOTH SL AND CL

We show our proposed attack, class-wise contrasting poisoning is effective on both SL and CL. In Table 4, we see that contrasting poisons, no matter generated by which CL framework, can always reduce the accuracy of SL to about 10%. (The dataset is CIFAR-10.) It can successfully disable the SL since its class-wise structure

Table 4: Attack efficacy on SL and CL.

| Attack Type + Attacker's Alg. | Victim's Algorithm | |
| --- | --- | --- |
| | Supervised | SimCLR |
| ADVERSARIAL POISONING | 8.7 | 81.5 |
| UNLEARNABLE EXAMPLES | 19.9 | 91.3 |
| CONTRASTIVE POISONING (C) (SIMCLR) | 10.2 | 68.0 |
| CONTRASTIVE POISONING (C) (MOCO) | 10.0 | 60.9 |
| CONTRASTIVE POISONING (C) (BYOL) | 10.1 | 60.7 |

shortcuts SL and makes the model associate the class with the noise attached instead of the content of images. The result is significant and means the victim cannot defend our attack even if he/she labels all the poisoned data and runs SL. The result also suggests that contrastive poisoning is the strongest indiscriminate poisoning attack. On the one hand, CP poisons SL as well as prior poisons against SL. On the other hand, CP is the only poison that is effective on CL.

## 5.5 ANALYSIS OF CONTRASTIVE POISONING

**(a) Attack efficacy and the fraction of poisoned data.** We evaluate attack efficacy as a function of the percentage of poisoned data, $p$. The experiments are conducted on CIFAR-10 with SimCLR. Table 5 reports the attack perfor-

Table 5: Accuracy of different poisoning ratios.

| Percentage Poisoning ($p$) | 100% | 90% | 80% | 50% | 20% | 10% | 5% |
| --- | --- | --- | --- | --- | --- | --- | --- |
| CLEAN ONLY (100%−$p$) | – | 70.6 | 78.6 | 85.2 | 87.9 | 89.2 | 89.5 |
| CONTRASTIVE POISONING (C) | 68.0 | 76.5 | 80.7 | 86.4 | 89.5 | 89.6 | 89.8 |
| CONTRASTIVE POISONING (S) | 44.9 | 59.1 | 72.2 | 83.7 | 88.1 | 88.6 | 89.6 |

mance as a function of the percentage of data that gets poisoned. For reference, we also show the accuracy if the victim model is trained using only the $100\% - p$ clean samples. The results show, for both class-wise and sample-wise contrastive poisoning, the attack power gradually decreases as the percentage of poisoned data decreases. This gradual decline is desirable and means that the attacker can control the impact of the poison by changing $p$. We also note that in comparison to past work on indiscriminate poisoning of SL, CP is much more resilient to the reduction in the fraction of poisoned data than past attacks whose poisoning effects quickly diminish when the data is not 100% poisoned (Shan et al., 2020).

**(b) Importance of the dual-branch scheme for CL algorithms with a momentum encoder.** We compare the accuracy resulting from a dual-branch scheme where the noise gradient is back-propagated through both the branch with and without a momentum encoder, and a single-branch scheme that does not back-propagate the noise gradient through the momentum branch. The experiment is conducted on CIFAR-10. We evaluate both types of contrastive poisoning. The attacker uses the same CL algorithm as the

Table 6: Attack results of poisoning generated with dual-branch gradient v.s. single-branch gradient.

| Attacker Algorithm | Victim Algorithm | | |
| --- | --- | --- | --- |
| | SimCLR | MoCo v2 | BYOL |
| CP (S) + MOCO V2 (SINGLE) | 65.9 | 74.8 | 87.2 |
| CP (S) + MOCO V2 (DUAL) | 54.9 (+11.0) | 55.1 (+19.7) | 71.1 (+16.1) |
| CP (C) + MOCO V2 (SINGLE) | 69.4 | 71.6 | 70.0 |
| CP (C) + MOCO V2 (DUAL) | 60.9 (+8.5) | 61.9 (+9.7) | 59.5 (+10.5) |
| CP (S) + BYOL (SINGLE) | 71.6 | 73.8 | 79.9 |
| CP (S) + BYOL (DUAL) | 65.1 (+6.5) | 64.2 (+9.6) | 59.6 (+20.3) |
| CP (C) + BYOL (SINGLE) | 68.3 | 70.4 | 66.7 |
| CP (C) + BYOL (DUAL) | 60.7 (+7.6) | 61.8 (+8.6) | 56.9 (+9.8) |

victim. As shown in Table 6, back-propagating the gradient along both branches unanimously improves the attack effectiveness and leads to 7% to 20% drop in accuracy over the single branch scheme. This is because back-propagating the gradient to the noise through both branches allows the noise to learn how to adversely change the output of the momentum encoder as well, instead of only attacking the encoder.

**(c) Impact of model architecture.** We generate contrastive poisons using ResNet-18 and SimCLR. The victim runs SimCLR on these poisoned datasets and tries different model architectures including VGG-19, ResNet-18, ResNet-50,

Table 7: Attack transferability across architectures.

| Attack Type | VGG-19 | ResNet-18 | ResNet-50 | DenseNet-121 | MobileNetV2 |
| --- | --- | --- | --- | --- | --- |
| NONE | 88.3 | 91.8 | 92.8 | 93.5 | 89.4 |
| CP (S) | 35.1 | 44.9 | 49.1 | 48.4 | 42.6 |
| CP (C) | 65.5 | 68.0 | 71.6 | 69.6 | 61.6 |

DenseNet-121, and MobileNetV2. The results are shown in Table 7. We can see that our attacks are effective across different backbone architectures.

**(d) Understanding CP noise.** In Figure 3, we visualize the noise generated by contrastive poisoning against SimCLR, and compare it to the noise generated by prior attacks against supervised learning. (A visualization of the MoCo and BYOL noise can be found in Appendix C.1.) We observe that the noise that poisons supervised learning has much simpler patterns than the noise that poisons SimCLR. Intuitively, this indicates that poisoning contrastive learning is harder than poisoning supervised

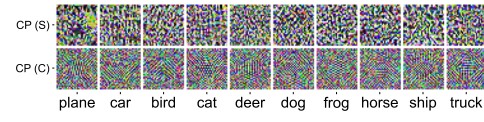

(a) Poisoning noise for supervised learning.    (b) Poisoning noise for SimCLR.

Figure 3: Visualization of the poisoning noise for supervised learning and contrastive learning. Note that some types of noise are sample-wise, we randomly sample one from each class to visualize.

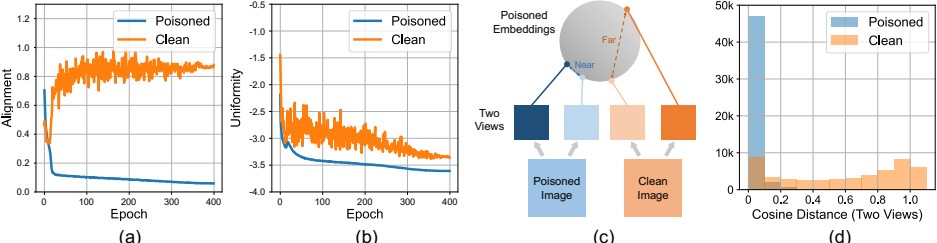

(a)    (b)    (c)    (d)

Figure 4: (a,b) Progression of the alignment and the uniformity loss during training. (c) Illustration of the poisoned encoder aligning poisoned image but not clean image. (d) Histograms of the cosine distance between the embeddings of two views of poisoned image (blue), clean image (orange).

learning. This is compatible with the results in Figure 1 and Table 1, which show that SL is more vulnerable to poisoning attacks than CL.

Next, we explain the working mechanism of the CP noise: (1) it does not work by shortcutting the classification task as all prior poisons did; (2) it works by shortcutting the task of aligning two augmented views of a single image. Specifically, prior work has observed that the noise that poisons supervised learning tends to cluster according to the original class labels as shown in Figure 5(a), and is linearly separable (Yu et al., 2022). We apply the same test on noise for attacking SimCLR and find it is not linearly separable, as shown in Figure 5(b). The reason for noise

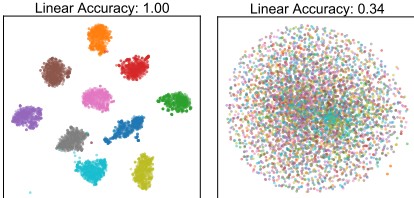

(a) noise attacks SL    (b) noise attacks CL

Figure 5: t-SNE of the poisoning noise.

for SL being linearly separable is that the supervised learning loss (i.e., CE) focuses on classification. Thus, to shortcut the SL loss, the attack inserts in each image noise that makes it easy to classify the images without looking at their content. Hence, the noise is clustered according to the classes.

In contrast, the CL loss (e.g., InfoNCE) is not focused on classification. The InfoNCE loss can be interpreted as a combination of alignment (aligning different views of the same image) and uniformity (spreading features) (Wang & Isola, 2020). To understand which component is shortcutted by the poisoning noise, we monitor the progression of their values during training. Figure 4(a,b) shows an example of running SimCLR on CIFAR-10 poisoned by class-wise CP. Blue curves are the loss computed on the poisoned dataset while the orange curves are the loss computed on the clean dataset, for comparison. We can see it is the alignment that is severely shortcutted as there is a huge gap between the poisoned alignment loss and the clean alignment loss. It means our poisons deceive the model to believe it aligns the two views of the same image while not achieving it, i.e., the cosine distance between features of two views of the poisoned image is small while that distance for the clean image is large (Figure 4(c)). We show the distance histograms in Figure 4(d). We can see the distance between poisoned features is always close to zero while the distance between clean features could be as large as one.

## 6 DEFENSES

**(a) Adversarial training.** Prior research on indiscriminate poisoning attacks against SL shows that adversarial training is the most effective countermeasure (Tao et al., 2021; Fowl et al., 2021a). Intuitively, it is because the poisoned data is in a small $l_\infty$ ball of the clean data. Adversarial robustness ensures that the model makes the same prediction within a small neighborhood around an input sample. Thus, its good accuracy on the poisoned training set can translate to the clean

test set. To enhance adversarial robustness of contrastive learning, several specialized adversarial training frameworks have been proposed: Kim et al. proposes to take adversarial examples of an anchor as its positive samples; Jiang et al. proposes to use a secondary encoder that is trained with adversarial examples and contrast it with a normally trained encoder. The current SOTA framework is AdvCL (Fan et al., 2021) which uses high-frequency image components and pseudo-supervision stimulus to augment contrastive adversarial training. In our experiment, we test the defense power of AdvCL against indiscriminate poisoning attacks on CL.

**(b) Data augmentation.** Data augmentations have also been extensively studied as defense mechanisms against poisoning attacks (Tao et al., 2021; Huang et al., 2021). We test three traditional data augmentations: *Random Noise*, which adds random white noise to the input; *Gauss Smooth* which applies a Gaussian filter to the input; and *Cutout* (DeVries & Taylor, 2017) which excises certain parts of the input. We further propose a new data augmentation based on *Matrix Completion*. The augmentation has two steps: first, it randomly drops pixels in the image; second, it reconstructs the missing pixels via matrix completion techniques (Chatterjee, 2015).

**(c) Experiments and results.** We conduct our experiments on CIFAR-10 using SimCLR and ResNet-18. For AdvCL, we use its default configurations on CIFAR-10. For data augmentations, we ablate their hyper-parameters. Specifically, for Random Noise, we control the standard deviation of the white noise to be small ($\sigma = 8/255$) or large ($\sigma = 64/255$). For Gauss-Smooth, we control the size of the Gaussian kernel to be small ($3 \times 3$) or large ($15 \times 15$). For Cutout, we follow a standard setting that

Table 8: Performance of various defenses.

| Defense Methods | CP (S) | CP (C) | Average |
|---|---|---|---|
| NO DEFENSE | 44.9 | 68.9 | 56.9 |
| RANDOM NOISE ($\sigma = 8/255$) | 54.1 | **90.3** | 72.2 |
| RANDOM NOISE ($\sigma = 64/255$) | 73.6 | 73.6 | 73.6 |
| GAUSS SMOOTH ($k = 3$) | 47.8 | 87.9 | 67.9 |
| GAUSS SMOOTH ($k = 15$) | 59.7 | 62.0 | 60.9 |
| CUTOUT | 47.7 | 75.0 | 61.4 |
| ADVERSARIAL TRAINING | 79.3 | 82.3 | 80.8 |
| MATRIX COMPLETION | **85.6** | 88.2 | **86.9** |
| CLEAN DATA | | 91.8 | |

excises a single hole of size $16 \times 16$. For Matrix-Completion, we adopt a pixel dropping probability of 0.25 and reconstruct the missing pixels using the universal singular value thresholding (USVT) algorithm (Chatterjee, 2015) with 50% of singular values clipped. In Appendix C.3, we visualize those augmentations.

Table 8 shows that among all defenses, only Adversarial Training and Matrix Completion can stably work under both attacks. Further, Matrix Completion gives the best defense accuracy. On average it achieves 86.9% accuracy which is 5.9% higher than the second best strategy, adversarial training. This is because adversarial training methods, although effective, have a limited performance upper-bound because they trade-off accuracy for increased robustness (Zhang et al., 2019), which means adversarial training unavoidably hurts the model's performance on clean data. Our results also reveal that sample-wise poison is more resilient to augmentation-based defenses than class-wise poison, which is highly sensitive to the defense mechanisms.

## 7 CONCLUDING REMARKS

This paper provides the first study of indiscriminate data poisoning attacks against contrastive learning (CL). It shows that prior indiscriminate poisoning attacks (which have targeted SL) can be overcome by using CL. To attack CL, it proposes a new, contrastive poisoning, and demonstrates its effectiveness. Furthermore, it shows that contrastive poisoning can poison both SL and CL. The paper also studies defense mechanisms and shows that data augmentation based on matrix completion is most effective against indiscriminate poisoning of CL.

We also highlight some limitations. In particular, the more transferable attack (i.e., class-wise CP) requires the attacker to have the labels. Having access to labels, however, is assumed in all prior indiscriminate poisoning attacks. Another potential limitation is the attacks typically require poisoning a relatively large percentage of data. Again this is a general limitation for indiscriminate poisoning of both SL and CL. In fact, our work reduces the required fraction of poisoned data. Also, in indiscriminate poisoning, the attacker is usually the dataset owner who wants to share the data while preventing others from extracting sensitive information. Many sensitive datasets have a small number of owners (e.g., a medical institution sharing pathology slides or x-ray images, a weather service sharing satellite images) in which case the owner can poison most or all of the shared data. We believe that these limitations do not hamper the value of the paper in providing the deep learning community with novel findings and a solid understanding of the growing area of data poisoning.

ACKNOWLEDGEMENTS

We are grateful to Peng Cao for reviewing and polishing the paper. We thank Yilun Xu and the members of the NETMIT group for reviewing the early drafts of this work and providing valuable feedback. We extend our appreciation to the anonymous reviewers/area chair for their insightful comments and constructive suggestions that greatly helped in improving the quality of the paper. Lastly, we acknowledge the generous support from the GIST-MIT program, which funded this project.

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

## A    DIFFERENTIABILITY OF DATA AUGMENTATIONS

In this section, we show that the image augmentations used in common contrastive learning methods (e.g., SimCLR, MoCo, BYOL) are differentiable. We categorize the augmentations into two groups: (1) geometric transformations that change the spatial locations of pixels; (2) color space transformations that modify the pixel values.

Let $I \in \mathbb{R}^{3 \times H \times W}$ denote the RGB input image of height $H$ and width $W$, $O \in \mathbb{R}^{3 \times P \times Q}$ denote the output image of height $P$ and width $Q$. $P, Q$ are not necessarily the same as $H, W$ since some augmentations may change the size of the image such as crop and resize.

**Geometric transformations** include cropping, resizing, and horizontal flipping, which can be considered as affine transformations.

- *Cropping*: Given the coordinate of top-left cropping location $(c_x, c_y)$ in the input image, the affine transformation matrix (in homogeneous coordinate) can be written as:

$$T = \begin{bmatrix} 1 & 0 & -c_x \\ 0 & 1 & -c_y \\ 0 & 0 & 1 \end{bmatrix}.$$

- *Resizing*: the affine transformation matrix can be written as:

$$T = \begin{bmatrix} P/H & 0 & 0 \\ 0 & Q/W & 0 \\ 0 & 0 & 1 \end{bmatrix}.$$

- *Horizontal flipping*: the affine transformation matrix can be written as:

$$T = \begin{bmatrix} 1 & 0 & 0 \\ 0 & -1 & W-1 \\ 0 & 0 & 1 \end{bmatrix}.$$

Given the affine transformation matrix $T$, the pixel value of location $(c, p, q)$ ($c \in \{0, 1, 2\}$, $p \in \{0, 1, \ldots, P-1\}$, $q \in \{0, 1, \ldots, Q-1\}$) in the output image can be computed as

$$O_{c,p,q} = \sum_{h,w} \alpha_{h,w} \cdot I_{c,h,w},$$

where for cropping and horizontal flipping, $\alpha_{h,w} = 1$ if $\begin{bmatrix} h \\ w \\ 1 \end{bmatrix} = T^{-1} \begin{bmatrix} p \\ q \\ 1 \end{bmatrix}$ otherwise $\alpha_{h,w} = 0$. For resizing, $\alpha_{h,w}$ depends on the interpolating methods. In general, $\alpha_{h,w}$ is the interpolation weighting factor if locations $(h, w)$ in the input image are the pixels to be used for interpolating the pixel value of location $(h', w')$ where $\begin{bmatrix} h' \\ w' \\ 1 \end{bmatrix} = T^{-1} \begin{bmatrix} p \\ q \\ 1 \end{bmatrix}$, otherwise $\alpha_{h,w} = 0$. Take the nearest neighbor interpolation as an example, $\alpha_{h,w} = 1$ if $(h, w)$ is the nearest to $(h', w') = (\frac{pH}{P}, \frac{qW}{Q})$ otherwise $\alpha_{h,w} = 0$. The above computation is differentiable since $\frac{\partial O_{c,p,q}}{\partial I_{c,h,w}} = \alpha_{h,w}$.

**Color space transformations** include grayscaling, brightness, contrast, and saturation adjustment. Those operations are pixel-wise and do not change the image size. Thus $H = P$ and $W = Q$.

- *RGB to grayscale*: $O_{c,p,q} = 0.299 \cdot I_{0,p,q} + 0.587 \cdot I_{1,p,q} + 0.114 \cdot I_{2,p,q}, \forall c, p, q$.
- *Brightness adjustment*: $O = r_b \cdot I$, where $r_b$ is the brightness adjustment factor.
- *Contrast adjustment*: $O = r_c \cdot I + (1 - r_c) \cdot \text{mean}(G) \cdot J_{3,H,W}$, where $r_c$ is the contrast adjustment factor, $G$ is the grayscaled image of $I$, and $J_{3,H,W}$ is all-ones tensor.
- *Saturation adjustment*: $O = r_s \cdot I + (1 - r_s) \cdot G$, where $r_s$ is the saturation adjustment factor and $G$ is the grayscaled image of $I$.
- *Hue adjustment*: $O = f(r_h \cdot h(I), s(I), v(I))$, where $r_h$ is the hue adjustment factor, $f$ is the HSV to RGB function, $h, s, v$ are functions mapping RGB to hue, saturation and value which are axes of HSV color space. Note that $f, h, s, v$ are all differentiable.

As can be seen from the above representations, the operations are all differentiable – i.e., all $\frac{\partial O_{c,p,q}}{\partial I_{c,h,w}}$ are determined by the above equations.

**Implementation.** We implement our own differentiable data augmentations by leveraging the design of an open-source differentiable computer vision library, Kornia (https://github.com/kornia/kornia). We do not directly use Kornia in our code since it uses an image processing backend inconsistent with PyTorch. Kornia uses OpenCV as the backend for image processing operations while PyTorch uses PIL. We have implemented our differentiable augmentations using PIL to make them consistent with PyTorch.

## B EXPERIMENT DETAILS

### B.1 DETAILS OF DATASET SPLIT

We follow the standard dataset split for contrastive learning.

- For CIFAR-10/100, we use 50K for both pre-training the encoder and training the linear classifier, and 10K to test the linear classifier.
- For ImageNet-100, we use ∼126.7K for training and 5K for testing.
- For STL-10, we only use it for linear probing. It has a standard split of 5K labeled trainset and 8K testset.

### B.2 DETAILS OF CONTRASTIVE LEARNING

Table 9: Hyper-parameters for different contrastive learning algorithms in our experiments.

|  | SimCLR | MoCo v2 | BYOL |
|---|---|---|---|
| Optimizer | SGD | SGD | SGD |
| Weight Decay | $10^{-4}$ | $10^{-4}$ | $10^{-4}$ |
| Learning Rate (LR) | 0.5 | 0.3 | 1.0 |
| LR Scheduler | Cosine | Cosine | Cosine |
| Encoder Momentum | - | 0.99 | 0.999 |
| Loss function | InfoNCE | InfoNCE | MSE |
| InfoNCE temperature | 0.5 | 0.2 | - |

Both the victim and attacker require training contrastive learning models. We follow the standard configurations for each contrastive learning framework Chen et al. (2020a); He et al. (2020); Grill et al. (2020). Table 9 lists the hyper-parameters specific to each framework. For CIFAR-10/-100, the models are trained for 1000 epochs with a batch size of 512. For ImageNet-100, the models are trained for 200 epochs with a batch size of 128.

**Linear probing.** When training the linear classifier to evaluate the learned representation, we use an SGD optimizer with a momentum of 0.9, a weight decay of 0, and an initial learning rate of 1.0 for CIFAR-10/-100 and 10.0 for ImageNet-100. The learning rate decays by a factor of 0.2 at epoch 60, 75, and 90. We set batch size to 512 and train the linear classifier for 100 epochs. Those linear probing hyper-parameters are the same for all CL algorithms.

### B.3 DETAILS OF CONTRASTIVE POISONING

On the attacker side, we co-learn the poisoning noise and a feature extractor as shown in Algorithm 1. For sample-wise CP, $T = 600$, $T_\theta = 100$, $T_\delta = 100$, $T_p = 5$. For class-wise CP, $T = 200$, $T_\theta = 20$, $T_\delta = 20$, $T_p = 1$. In both attacks, we set the PGD learning rate $\eta_\delta$ to one tenth of the radius of the $L_\infty$ ball, i.e., $\eta_\delta = \epsilon/10 = 0.8/255$. In general, the principles are sample-wise CP needs more updates, i.e. more total updating rounds $T$ and more PGD steps $T_p$ since each sample has its own perturbation which gives a lot of parameters to learn. On the other hand, class-wise CP needs more frequent alternating, i.e. smaller $T_\theta$ and $T_\delta$. It is because, for class-wise perturbation, each of them will be updated by many samples in one batch. We do not want to over-optimize them such that

they over-fit the model $f_\theta$ at that time point instead of focusing on the whole dynamics of the model evolving.

### B.4    COMPUTING RESOURCES CONSUMPTION

**Hardwares.** We run CIFAR-10/-100 experiments on 4 NVIDIA TITAN Xp GPUs. We run ImageNet-100 experiments on 4 NVIDIA Tesla V100 GPUs.

**Running time for CIFAR-10/-100.** (1) On the attacker side, generating noise using contrastive poisoning takes 8-12 hours. (2) On the victim side, training a contrastive learning model (e.g., SimCLR, MoCo, BYOL) takes 8-10 hours.

**Running time for ImageNet-100.** (1) On the attacker side, generating noise using contrastive poisoning takes 20-30 hours. (2) On the victim side, training a contrastive learning model (e.g., SimCLR) takes 15-20 hours.

## C    ADDITIONAL VISUALIZATIONS

### C.1    POISONING NOISE

Figure 6 visualizes the poisoning noise against MoCo v2 and BYOL. We can see that those noises learned from MoCo v2, BYOL, as well as SimCLR (shown in Figure 3), share similar structures. Especially, the class-wise error-minimizing poisoning (EMP-CL-C) learned from three contrastive learning frameworks has similar chessboard-like patterns. This may explain why EMP-CL-C transfers very well on attacking different CL frameworks.

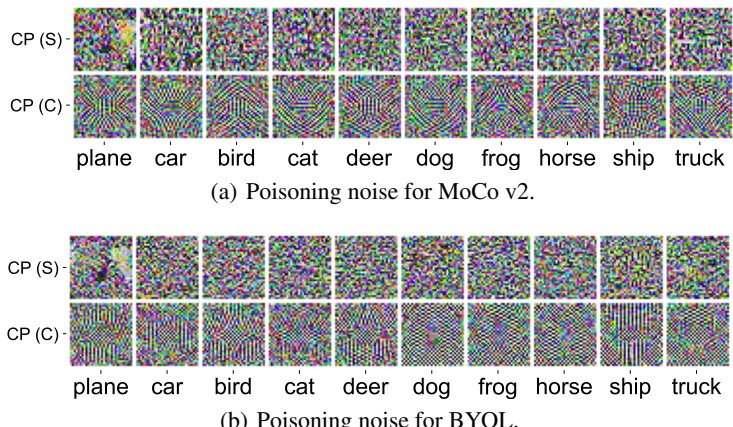

(a) Poisoning noise for MoCo v2.

(b) Poisoning noise for BYOL.

Figure 6: Visualization of the poisoning noise for MoCo v2 and BYOL. Note that for the sample-wise noise, we randomly sample one from each class to visualize.

### C.2    ANALYSIS OF SAMPLE-WISE CONTRASTIVE POISONING NOISE

In the main paper (Figure 4), we show the progression of the alignment loss and the uniformity loss when training SimCLR on CIFAR-10 poisoned by class-wise contrastive poisoning, as well as the histogram of the cosine distance of two views of the poisoned data in the embedding space.

We apply the same test to sample-wise contrastive poisoning and get similar results as class-wise contrastive poisoning. Figure 7 shows the loss progressions of sample-wise contrastive poisoning. The curves show the InfoNCE loss and the alignment loss are severely shortcutted. Figure 8 shows the distance histograms of the embeddings of the two views of the image poisoned by sample-wise CP (in blue). We can see the distance of poisoned features is always close to zero.

Further, comparing the loss curves between sample-wise CP and class-wise CP, we can find that sample-wise CP leads to a larger gap between the poisoned and clean alignment losses than class-wise CP. It indicates that sample-wise CP has a stronger shortcutting effect. This is consistent with the fact that sample-wise CP has the highest attacking efficacy on SimCLR, as shown in Table 3.

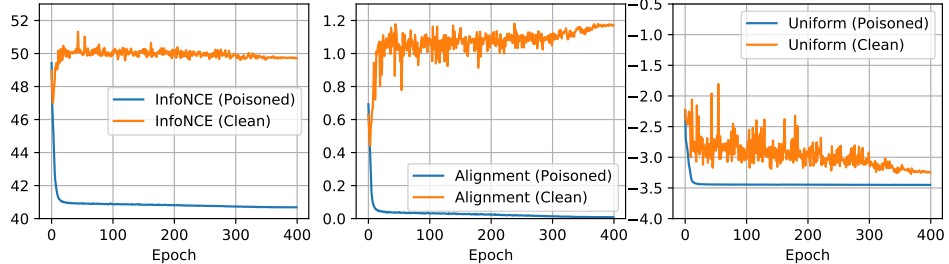

Figure 7: Loss progression of SimCLR on CIFAR-10 poisoned by sample-wise CP.

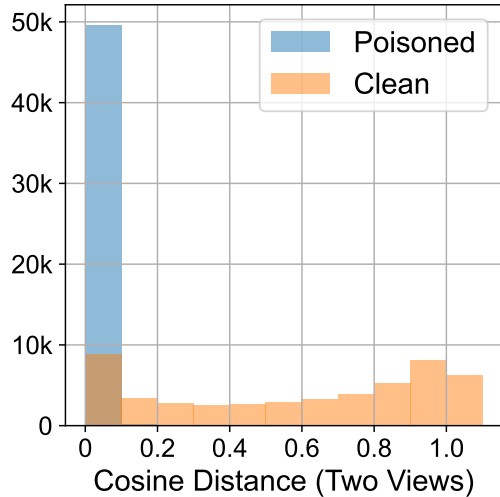

Figure 8: Histograms of the cosine distance between the embeddings of two views of image poisoned by sample-wise CP (blue), clean image (orange).

## C.3    DATA AUGMENTATION FOR DEFENSE

Figure 9 visualizes the data augmentations that we use to defend against indiscriminate poisoning of contrastive learning for ten randomly selected CIFAR-10 images. In the figure, the order of augmentations is the same as the order in Table 8.

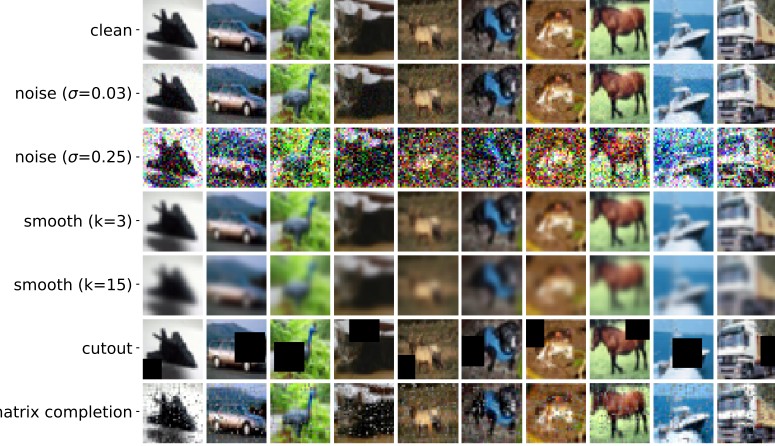

Figure 9: Visualization of the data augmentations (introduced in Section 6) for defending against indiscriminate poisoning attacks.

