# OpenReview forum: "Indiscriminate Poisoning Attacks on Unsupervised Contrastive Learning"
_ICLR.cc/2023/Conference — ICLR 2023 notable top 25%_

### Official Review · Reviewer_vT93 · 2022-10-14

**Confidence:** 4
**Correctness:** 3
**Technical Novelty And Significance:** 2
**Empirical Novelty And Significance:** 2
**Recommendation:** 6

**Clarity, Quality, Novelty And Reproducibility:**

Please see Strength And Weaknesses. I hope the authors would reply to my first two concerns to further improve this work.

The paper is well written in general. There is a missing reference in Appendix C.2.

**Strength And Weaknesses:**

**Strength**

1. This paper gives the first attack against contrastive learning. It is interesting to know that one can find shortcuts for the instance-wise CL task.

2. Several interesting tricks, e.g., letting the gradient back through the momentum encoder and differentiable data augmentations, are proposed to improve the attack.

3. The authors show several defenses, including a newly proposed matrix completion, that can be used to defend against contrastive poisoning.

**Weaknesses**

1. How important is using differentiable data augmentations? I do not see an ablation study on this. Will the poisons be useless without this trick?

2. In Figure 4 you explain why contrastive poisoning works but only use class-wise poisons. It would be better if you could include the results of sample-wise poisons.

3. Most of the experiments assume the attacker has full access to the training data. Although most of previous works also use this assumption, it does not change the fact that it would probably be unrealistic.

4. The explanation of why CP works is that it provides shortcuts for the model to produce similar representations of different augmented views. If this is true, it is possible to directly optimize the poison so that it remains unchanged when the data augmentation is applied. For example, a circle is invariant to rotation. This data/model independent approach is more efficient than the optimization procedure.


**Summary Of The Paper:**

This paper poisons unlabeled training data of contrastive learning (CL) to reduce the test accuracy of linear probing. The imperceptible poisons are iteratively optimized to minimize the CL objective function. After the poisons are generated, the authors add them to the training set and run the target CL algorithm from scratch. When the entire training set is poisoned, there is a clear drop in the linear probing accuracy, e.g., from ~90% to ~50% on CIFAR-10.


The poison generation proceeds as follows. At each iteration, the model parameters are first updated to minimize CL loss and then the poisons are also updated to minimize CL loss. This follows the basic idea in Huang et al (2021) although Huang et al (2021) minimize classification instead of CL loss. One worth mentioning difference is the authors show data augmentations of common CL algorithms are differentiable. They take the gradients of data augmentation into consideration when updating the poisons.

The authors analyze why the poisons work. They show the poisons work by shortcutting the instance-wise CL task, whose objective is to produce similar representations of two different augmented views. They show the poisoned embeddings of two augmented views are close while the clean embeddings are far.


**Summary Of The Review:**

Given that this paper is the first to show it is possible to design shortcuts for instance-wise CL task, I recommend acceptance.

---

> ### Author Response · Authors · 2022-11-16
> **Response to Reviewer vT93**
>
> We thank the reviewer for the constructive reviews. Below, we answer the specific questions. We hope our response addresses the reviewer's concerns, especially the first two. We are more than happy to answer any follow-up questions.
>
> ---
>
> **Q1. How important is using differentiable data augmentation?**
>
> The poisons will have zero effect if one does not differentiate the data augmentation. Below, we provide an ablation study. To disable differentiable data augmentation, we take the gradient of the augmented views (which do not backpropagate through the data augmentation ) to optimize the poisons. We do an experiment on CIFAR-10 while the attacker and the victim use SimCLR and ResNet18. We test the attacking efficacy of both class-wise and sample-wise CP learned without differentiable data augmentation. As shown in the table below, the linear-probing accuracy of the poisoned encoder is 90.0% and 88.9% without differentiable data augmentation. This means that differentiating the data augmentation is critical to the success of contrastive poisoning.
>
> |   Attack     | Differentiable Data Aug (ours)| No Differentiable Data Aug|
> |:------:|:-----------------------------:|:-------------------------:|
> | CP (C) |    68.0 	                   |    90.0   			   |
> | CP (S) |    44.9   	                   |    88.9                   |
>
> ---
>
> **Q2. Explain the working mechanism of sample-wise poisons.**
>
> Thanks for the suggestion. We include the results of sample-wise CP in Appendix C.2. Specifically, the loss progression curves for sample-wise CP have already been included in our original submission. In our revision, we further include the embedding cosine distance histogram of the two views of the poisoned image for sample-wise CP (in Figure 8).
>
> As the results show, sample-wise poisons work the same way as class-wise poisons. They have similar progression curves of the alignment and the uniformity loss as well as similar histograms of the cosine distance.
>
> ---
>
> **Q3. Assumption of full access to the training data.**
>
> We are aware that the assumption of accessing full training data is a limitation. However, as we discussed in the conclusion: this is a general limitation for indiscriminate poisoning of both SL and CL. In fact, our work reduces the required fraction of poisoned data by showing that our contrastive poisoning remains certain attacking efficacy when attackers only have access to a partial training set (Table 5). Also, in indiscriminate poisoning, the attacker could be the dataset owner who wants to share the data while preventing others from extracting sensitive information. Many sensitive datasets have a small number of owners (e.g., a medical institution sharing pathology slides or x-ray images, a weather service sharing satellite images) in which case the owner can poison most or all of the shared data. We believe that this limitation does not hamper the value of the paper in providing the community with novel findings and a solid understanding of the growing area of data poisoning.
>
> ---
>
> **Q4. Directly learning poisons by shortcutting alignment loss.**
>
> It is an interesting idea to directly learn poison to be invariant to the data augmentations. We also think of this in our study. However, unfortunately, we find this idea is not applicable due to the fact that contrastive learning uses very strong augmentations such that it is impossible to have noise that is invariant to them. Specifically, one augmentation (in color-jitter) is brightness adjustment which multiplies the RGB channel by a random scalar. So if a pixel is unchanged after being multiplied by a non-zero scalar, the pixel value has to be zero. It means noise that is variant to random color-jitter operation can only be all zeros.
>
> Further, we would like to emphasize the necessity of having a data/model-dependent poisoning procedure as our CP does. First, we note that embeddings of *poisons* alone being invariant under augmentations are different from embeddings of *poisons plus images* being invariant under augmentations. Effective attacks require the latter instead of the former one. Thus it is necessary to do data-dependent optimization to make sure that poisoned images (instead of poisons themselves) shortcut the alignment loss. Second, we note that shortcutting one fixed CL pretrained model is not enough for the attack. Because during contrastive learning, the model is evolving. It is important for the poisons to shortcut the model at every timepoint during training. Thus doing a model-dependent poisoning attack by co-learn poisons and a model is more effective.
>
> ---
>
> **Typos**
>
> Thanks for spotting the missing reference in Appendix C.2. We have fixed it in our revision.

---

> > ### Comment · Reviewer_vT93 · 2022-11-24
> > **Reply to author response**
> >
> >
> > Thank you for the new experiments. I have no further questions and maintain my positive recommendation.

---

> > > ### Author Response · Authors · 2022-11-24
> > > **Thank you for your reply**
> > >
> > > We thank the reviewer for the reply.  We are glad to hear that the reviewer stays positive. We would really appreciate it if the reviewer could consider raising the score, given that our response adequately solves the reviewer’s concerns. Please feel free to let us know if there are any further questions. We thank the reviewer again for the time and constructive feedback.

---

### Official Review · Reviewer_Cwex · 2022-10-25

**Confidence:** 5
**Correctness:** 4
**Technical Novelty And Significance:** 3
**Empirical Novelty And Significance:** 4
**Recommendation:** 6

**Clarity, Quality, Novelty And Reproducibility:**

The paper is clear in describing settings, technical details and empirical finding. Its novelty and impact is somewhat hindered by the less realistic attack setting.

**Strength And Weaknesses:**

**Strength**

This paper is technically sound. The experiments considers a wide range of baselines. I appreciate the effort.

**Weaknesses**

As the authors have self-identified in the limitation, the attack setting seems to be slightly artificial: the attacker has control over *all* training data is a very strong assumption... The attack mechanism is not too surprisingly novel either. I feel that the paper has presented much empirical evidence of what effect the attack is able to bring, but fails to provide in-depth analysis of its implication. For example, on Page 8, the authors have shown that noise in SL attack is sometimes linear separable, while that in CL attack is not. It is a very interesting observation. However, does that mean linear separable noise is easier to detect? or, what does this linear (non)separability suggest? In future edition or work, the authors may dig further the implication of such phenomena to contrastive learning or more general learning problem. Such insights could be more helpful than the attack/defense itself, given that the adversarial setting is not very common in practice.



**Summary Of The Paper:**

This paper considers an adversarial setting for contrastive learning: the attacker is allowed to add imperceptible perturbation to all training samples of CL, which will reduce CL's effectiveness as a feature extractor. Experiments show that downstream task's performance will be lowered by the poisoning.

**Summary Of The Review:**

Overall, this is a paper with solid empirical evaluation but slightly lacking novelty. To me, it is on the borderline.

---

> ### Author Response · Authors · 2022-11-16
> **Response to Reviewer Cwex**
>
> We thank the reviewer for the comments. We are glad that the reviewer finds our work technically sound and clear in describing settings, technical details, and empirical findings.
>
> The major concern of the reviewer is with regard to the standard setting of indiscriminate poisoning attacks where the attacker is assumed to have control over all training data. We hope the reviewer does not penalize our work due to this standard setting in the area of indiscriminate poisoning attacks. First, this assumption by all prior works on the topic [1,2,3]. To make a fair comparison to prior works, we need to study this setting. Second, in contrast to prior works, we have also studied the settings where not all training data is poisoned (Please see Table 5). Third, we do believe there are real settings where the attacker has control of the full datasets. In particular,  some specialized and sensitive datasets can come from one source (e.g., a medical institution with patient data (e.g., [A,B,C]) or a weather service with satellite data (e.g.,[D,E]). The source may want to share this data with one or more entities but without allowing them to use the data for potentially unauthorized learning tasks. In this case, the source has control of the shared data and can use the attack to avoid unauthorized learning from the shared data.
>
> So we hope that the reviewer will reconsider as we are just following the standards in this sub-area (and in fact, if anything our work allows for significantly loosening this assumption).
>
>
> The reviewer also complains that *“... the paper has presented much empirical evidence of what effect the attack is able to bring, but fails to provide in-depth analysis of its implication.”* We note that we are the first to study indiscriminate poisoning attacks on unsupervised contrastive learning. Considering no prior work has studied indiscriminate attacks on CL, it is important to first deliver a successful attack with strong and extensive empirical results. We also have provided a high-level analysis of the results.  We believe the work is important since all prior indiscriminate poisoning attacks (which have targeted SL) can be overcome by using CL. It is common in our field to publish empirical result-driven work [1,2]. We hope the reviewer considers our work in a similar perspective to prior work in this field with its level of empirical evidence and analysis of implications.
>
> Reference:
> - [1] Adversarial examples make strong poisons. NeurIPS’21
> - [2] Unlearnable examples: Making personal data unexploitable. ICLR’21
> - [3] Availability attacks create shortcuts. KDD’22
>
> Dataset:
> - [A] “The Sleep Heart Health Study: design, rationale, and methods”, dataset owner: University of Arizona
> - [B] “Associations between sleep architecture and sleep-disordered breathing and cognition in older community-dwelling men: the Osteoporotic Fractures in Men Sleep Study”, dataset owner: California Pacific Medical Center Research Institute
> - [C] “Sleep-disordered breathing and cognition in older women”, dataset owner: University of California, San Francisco
> - [D] “Agriculture-Vision: A Large Aerial Image Database for Agricultural Pattern Analysis”, dataset owner: UIUC, Intelinair.
> - [E] “FloodNet: A High Resolution Aerial Imagery Dataset for Post Flood Scene Understanding”, dataset owner: University of Maryland

---

> ### Author Response · Authors · 2022-11-24
> **Thank you for your update**
>
> We thank the reviewer for updating the review and the evaluation.  If the reviewer has any further questions, we are more than happy to provide additional clarifications or experiments.

---

### Official Review · Reviewer_gmqR · 2022-10-25

**Confidence:** 4
**Clarity, Quality, Novelty And Reproducibility:** The paper is clearly written, well-or…
**Correctness:** 4
**Technical Novelty And Significance:** 3
**Empirical Novelty And Significance:** 3
**Recommendation:** 8

**Strength And Weaknesses:**

Strengths:

- This paper is the first to study availability attacks against unsupervised contrastive learning.
- A dual branch scheme is proposed to mitigate the optimization obstacles and thus boost the attack performance.
- Experiments provide several novel observations:
  - It is an interesting observation that sample-wise CP mostly performs better, while class-wise CP can transfer better.
  - While almost all previous poisons in this area are linearly separable, the proposed poison is not. This would make availability attacks more undetectable.
  - A very simple data augmentation method based on Matrix Completion is proposed to mitigate poisons. The authors find that this simple method can perform better than adversarial training.

Weaknesses:

- Given that Matrix Completion is very effective in defending against the proposed attack, it is natural then to ask whether this simple method can defend against previous poisons, such as unlearnable examples and adversarial poisoning.
- Table 8 tests the defense power of AdvCL (i.e. a type of adversarial training) against the proposed poisons. Is the perturbation radius of adversarial training $\epsilon=8/255$? Prior work (Tao et al., 2021, Figure 5) has shown that adversarial training with small $\epsilon$ could perform better. Is this also true in the context of unsupervised contrastive learning? By the way, adversarial training was first identified by Tao et al as a promising defense against those poisons. Thus, the last sentence of the introduction needs to polishing.
- There are availability attacks that can work by injecting dirty labels into the training data (Biggio et al., 2012). That is to say, not all availability attacks are imperceptible. Thus, the first sentence of the introduction needs to polishing.

**Summary Of The Paper:**

This paper studies clean-label availability attacks against unsupervised contrastive learning. The authors first observe that previous poisons, designed to compromise supervised learning, fail to harm contrastive learning algorithms, such as SimCLR and BYOL. Then, the authors propose a new attack method tailored to compromise contrastive learning. Surprisingly, the proposed attack is not only effective for contrastive learning, but also for supervised learning.

**Summary Of The Review:**

The paper explains why previous poisons fail and shows how to effectively compromise contrastive learning. This paper observes many novel empirical findings. These constitute a thorough study of clean-label availability attacks against unsupervised contrastive learning.

---

> ### Author Response · Authors · 2022-11-16
> **Response to Reviewer gmqR**
>
> We thank the reviewer for the thoughtful comments. We appreciate that the reviewer finds our paper has many novel empirical findings. Below, we respond to the comments in the review one by one.
>
> ---
>
> **Q1. Can Matrix Completion defend previous poisons?**
>
> Thanks for your suggestion. We test Matrix Completion’s defense efficacy against previous poisons. As shown in the table below, Matrix Completion has a strong defense power against Adversarial Poisons (boost accuracy from 8.7% to 77.4%) while has a very weak defense power against Unlearnable Examples (boost accuracy from 19.9% to 28.1%).
>
> We believe such a difference is caused by the structure of poisons generated by Adversarial Poisons (AP) and Unlearnable Examples (UE). As shown in Figure 3 in our paper, AP generates poisons that have many high-frequency components as our contrastive poisons. In contrast, UE generates poisons with few high-frequency components. Matrix Completion, by its nature of using SVD, is effective at breaking high-frequency poisoning noise and less effective at removing low-frequency poisoning perturbations.
>
> | Defense \ Attack  | Unlearnable Examples | Adversarial Poisons |
> |-------------------|:--------------------:|---------------------|
> | None              |         19.90        |          8.69       |
> | Matrix Completion |         28.12        |         77.38       |
>
> ---
>
> **Q2. Will adversarial training with smaller $\epsilon$ perform better defense against poisons on unsupervised contrastive learning?**
>
> For the AdvCL result in Table 8, we use the perturbation radius $\epsilon=8/255$. The original AdvCL paper [1] also only uses $\epsilon=8/255$. From the prior work provided by the reviewer (Tao et al., 2021, Figure 5), the accuracy curve is relatively flat as long as the perturbation radius is in a reasonable range.  So we believe using a relatively smaller $\epsilon$ will not change the accuracy much.
>
> ---
>
> **Polishing text**
>
> - We change the citation to “(Tao et al., 2021)” in the last sentence of the introduction. Thanks for spotting our misreference.
> - We removed the word “imperceptible” in the first sentence of the introduction. We changed the sentence to “Indiscriminate poisoning attacks are a particular type of data poisoning in which the attacker adds to the training data or labels perturbations that do not target a particular class, but lead to arbitrarily bad accuracy on unseen test data.” Hope it is more precise now. Thanks for the suggestion.
>
> ---
>
> We really appreciate the reviewer for providing us with constructive suggestions. We hope our response addresses the reviewer’s questions, and our revision makes our submission stronger.  We are more than happy to answer any follow-up questions.

---

> > ### Comment · Reviewer_gmqR · 2022-12-02
> > **Thanks**
> >
> > Thanks for the response. My concerns are well addressed. It is interesting to observe that matrix completion as a defense can largely mitigate adversarial poisons, while it is less effective in defending against unlearnable examples. This would inspire future work to study this method. I'd like to raise my score from 6 to 8.

---

> ### Author Response · Authors · 2022-11-24
> **Thank you**
>
> We thank the reviewer again for the constructive review. We hope our response and revision adequately address the reviewer’s concerns. We would really appreciate it if the reviewer could consider raising the score after evaluating our updates. Please feel free to let us know if there are other clarifications or experiments we can offer.

---

### Decision · Program_Chairs · 2023-01-20

**Decision:**

Accept: notable-top-25%

**Justification For Why Not Higher Score:**

The paper provides novel and interesting contributions worthy of a spotlight presentation. However these might not groundbreaking enough to warrant an oral presentation.

**Justification For Why Not Lower Score:**

This paper is the first to study indiscriminate poisoning on contrastive learning.
Given the increasing use of contrastive learning and the rich insights provided by this paper, the contributions are worthy of a spotlight presentation.

**Metareview: Summary, Strengths And Weaknesses:**

The paper studies indiscriminate poisoning attacks on contrastive learning. Existing attack strategies developed for supervised learning are shown to be ineffective on contrastive learning. A novel strategy is thus proposed that not only is shown to be highly effective on contrastive learning but also impacts supervised learning models. A new defense measure is proposed based on data augmentation.

All reviewers and the AC agree that this paper makes several interesting and novel contributions. As contrastive learning becomes increasingly popular in many fields, it is very important to study how vulnerable various approaches are and to understand how various attacks could harm the resulting models. The paper shed light on several of theses aspects and the authors did a good job at addressing the reviewers comments.

The weakness of the paper are adequately discussed by the authors and we strongly encourage them to pursue this line of work and perhaps design poisoning attacks that are even more efficient w.r.t to the fraction of poisoned data.


**Note From Pc:**

if the above contains the word "oral" or "spotlight" please see: "oral" presentation means -> notable-top-5% and "spotlight" means -> notable-top-25%. As stated in our emails, we are disassociating presentation type from AC recommendations